# Individual recognition and the 'face inversion effect' in medaka fish (*Oryzias latipes*)

Mu-Yun Wang[1,2]*, Hideaki Takeuchi[1,2]

[1]Department of Biological Sciences, Graduate School of Science, The University of Tokyo, Tokyo, Japan; [2]The Graduate School of Natural Science and Technology, Okayama University, Okayama, Japan

**Abstract** Individual recognition (IR) is essential for maintaining various social interactions in a group, and face recognition is one of the most specialised cognitive abilities in IR. We used both a mating preference system and an electric shock conditioning experiment to test IR ability in medaka, and found that signals near the face are important. Medaka required more time to discriminate vertically inverted faces, but not horizontally shifted faces or inverted non-face objects. The ability may be comparable to the classic 'face inversion effect' in humans and some other mammals. Extra patterns added to the face also did not influence the IR. These findings suggest the possibility that the process of face recognition may differ from that used for other objects. The complex form of recognition may promote specific processing adaptations, although the mechanisms and neurological bases might differ in mammals and medaka. The ability to recognise other individuals is important for shaping animal societies.

*For correspondence:
muyunwang@gmail.com

**Competing interests:** The authors declare that no competing interests exist.

## Introduction

In a social group, the ability to recognise other individuals correctly is essential for maintaining various social interactions in animals, such as pair-bonding, hierarchy, inbreeding avoidance, and recognition of offspring, nest mates, or neighbours (*Tibbetts and Dale, 2007*; *Wiley, 2013*). For example, some territorial birds can remember specific neighbours for a long period of time (*Godard, 1991*), and king penguins can identify their chick from thousands of conspecifics (*Aubin and Jouventin, 1998*). Receivers associate different types of identity signals, such as odour, sound, tactile, motion, electric or morphological cues, with certain individuals (*Sherman et al., 2009*) and identify them afterwards when necessary. In addition to looking at how animals recognise conspecifics, their mental representations of specific individuals can also give hints that allow us to judge their cognitive abilities. For example, hamsters have various odours for different body parts, and an unfamiliar hamster will categorise them as multiple individuals, while a previously interacted hamster can associate the odours to the specific individual (*Johnston and Bullock, 2001*). Animals may have complicated mechanisms to link multiple identity signals to different types of fitness-related tasks, or may use simpler rules to remember an individual. Among all of the individual recognition (IR) systems, face recognition is one of the most specific abilities, and is reported in animals from a number of distinct evolutionary lineages (*Kendrick and Baldwin, 1987*; *McKone et al., 2007*; *Van der Velden et al., 2008*, *Coulon et al., 2009*; *Racca et al., 2010*; *Sheehan and Tibbetts, 2011*). How faces are recognised, and whether the processes involved differ from those used to perceive other objects, is a main topic of interest in the field of cognitive psychology and biology.

In humans and some other mammals, faces are specially processed in cognitive, developmental and functional ways (*Calder, 2011*). Human infants are hypothesised to be attracted to faces

**eLife digest** Being able to recognize each other is crucial for social interactions in humans, as well as many other animals. To humans, faces are the most important body part to differentiate between one another. Humans read the face as a whole, rather than look at parts of the face, which is why it is harder to recognise a face when we see it upside-down, but not when we see an upside-down object.

Some other mammals also identify each other by the face and take longer to recognise an upside-down face, but this ability has never been observed in animals other than mammals. Previous research has shown that some fish species can distinguish between individuals. For example, female medaka fish prefer males they have seen before to 'strangers'. However, until now, it was not known if they can recognize individual faces, nor how they distinguish a specific male from many others.

To see if medaka fish use vision, smell or both cues to recognise mates, Wang and Takeuchi familiarised the fish before the mating test in different settings. In the first group, the male and the female could see each other but were kept in different tanks; in the second group to test odour cues, the male and the female were in the same tank but could not see each other; in the third group, the fish were in the same tank and could see each other; the fish in the fourth group were kept in different tanks and could not see each other. To make sure the fish can recognise and distinguish between fish or objects, Wang and Takeuchi also performed negative conditioning experiments, in which the females had to learn to form an association between a negative stimulus and a specific situation.

Wang and Takeuchi found that medaka fish use both vision and smell to distinguish between other fish, but could recognise each other based on vision alone. More specifically, the fish looked at the faces to tell others apart, and even when spots were added to their faces, the fish could still recognise the other. The mekada fish were also able to discriminate between two fish and two objects, but failed the task when the fish images were presented upside-down. However, when two objects were inverted, they were still able to tell the difference. This suggests that just like humans, faces may be special for fish too.

This is the first study that shows the face inversion effect in animals other than mammals. A next step will be to compare the different mechanisms between species, and identify the underlying genes and nerve cells responsible for face recognition. This will enable us to better understand social interactions in fish, and enhance our knowledge of how our own ability to recognize faces has changed from an evolutionary point of view.

innately (*Morton and Johnson, 1991*), but also develop face recognition skills and specific brain regions for processing faces during childhood. A familiar face can be individuated in 250 ms (*Jacques and Rossion, 2006*), and we can possibly remember more faces than other visual stimuli with similar variations in details and features. Studies of a neuropsychological disorder known as prosopagnosia or face blindness, in which individuals are unable to recognise faces but have no difficulty in recognising individuals by other modalities (such as voice) or in discriminating non-face objects (*Meadows, 1974*; *Behrmann et al., 2005*), have shown that facial recognition proceeds through specific cognitive and neural pathways (*Valentine, 1988*; *McKone et al., 2007*). In addition, the increase in recognition difficulty associated with inversion of faces is greater than that for the inversion of other types of visual stimuli (*Yin, 1969*). The so-called face-inversion effect indirectly indicates that faces are perceived configurally rather than only by specific features (such as the eyes, nose, or mouth), and that once inverted, such a global configuration is difficult to match and passes through routes which are used for recognising other objects (*Bartlett and Searcy, 1993*; *Haxby et al., 1999*; *Boutsen et al., 2006*). Likewise, the Thatcher illusion found in both humans (*Thompson, 1980*) and monkeys (*Adachi et al., 2009*; *Dahl et al., 2010*), in which the eyes and mouth are inverted relative to the face, becomes difficult to detect when upside down, further demonstrating that configural perception is interrupted when orientation is inverted.

Some other animals, ranging from mammals, birds and fish to invertebrates, have also been reported to use faces for IR (*Brown and Dooling, 1992*; *Kendrick et al., 1995*; *Bovet and*

*Vauclair, 2000*; *Van der Velden et al., 2008*, *Kohda et al., 2015*; *Parr and Hecht, 2011*; *Tibbetts, 2002*). Scientists have long argued that the face-specific processes are unique to humans or shared only by quite closely related species (*Tate et al., 2006*). However, such specialised ability may also have evolved in distinct animal taxa when selection force associated with complicated, repeated social interactions strongly favours IR.

The face inversion effect is the method most widely used in animals to test whether faces may be processed specifically, and researchers have identified this ability in some non-human primates (*Overman and Doty, 1982*; *Tomonaga, 1994*; *Parr et al., 1998*; *Vermeire and Hamilton, 1998*; *Neiworth et al., 2007*) and in sheep (*Kendrick et al., 1996*). Some monkeys failed to show such oriented-specific face-processing (*Rosenfeld and Van Hoesen, 1979*; *Bruce, 1982*; *Dittrich, 1990*; *Parr et al., 1999*; *Weiss et al., 2001*; *Gothard et al., 2004*), but many studies lacked the use of non-face signals as controls, making it difficult to interpret the results (*Parr et al., 1999*). Specialised neural systems for face recognition have been found in some non-human primates and in sheep (*Kendrick and Baldwin, 1987*; *Kanwisher and Yovel, 2006*; *Tsao et al., 2006*), providing great opportunities to interpret how these animals perceive faces perceptually and mechanically for comparative research. Other than the inversion effect, sheep, chimpanzees, and wasps exhibit better discrimination of conspecific faces than of non-face objects (*Kendrick et al., 1996*; *Parr et al., 1998*; *Sheehan and Tibbetts, 2011*). The difference between decision speed and accuracy in discriminating faces and non-facial stimuli is hypothesized to be due to face-specific perception (*Sheehan and Tibbetts, 2011*).

In the present study, we used a popular freshwater animal model, the medaka fish (*Figure 1A*), to test IR ability and to examine whether these animals perceive faces differently from non-face stimuli. Researchers have only recently found that fish can use facial pattern to individuate others. Manipulation using digital models demonstrated that two species of cichlid fish use facial patterns, but not body colouration, to recognise familiar individuals (*Kohda et al., 2015*; *Satoh et al., 2016*). A species of reef fish uses UV patterns on the face for species recognition, but there is no evidence of IR (*Siebeck et al., 2010*). Medaka are shoaling fish with diverse social behaviours that has become a popular model in genetic and neural research. Medaka females prefer males with larger body sizes (*Howard et al., 1998*) and longer fins (*Fujimoto et al., 2014*), or familiar males. Visual contact for 5 hr can shorten the time to mate for a pair of medaka, and a certain extrahypothalamic neuromodulatory system alters the preference in response to familiarity (*Okuyama et al., 2014*). Nonetheless, the cues used for medaka IR and the cognitive basis that underlies IR remain unknown. Here, we investigated the identity signals used for medaka IR, and whether the process of recognising other individuals differs from that for other objects. We propose that medaka can become a powerful model for understanding IR systems for many reasons. First, abundant closely related species with different social behaviours are available, allowing us to test the evolutionary background that promotes strict IR. Second, the social behaviours within the species are also variable. Medaka from different geographic regions or different inbred strains behave uniquely (*Tsuboko et al., 2014*), allowing us to investigate how ecological factors influence the use of identity signals, as well as the mechanisms behind these signals. Moreover, rich genetic techniques such as genome editing and epigenetic methods are available for medaka (*Kirchmaier et al., 2015*), providing powerful tools with which to solve complex questions.

The first aim of this study was to identify the cues used for medaka IR. We tested whether visual and odour cues are part of the identity signals, and whether the cues work collaboratively. We also investigated which visual components (such as appearance, motion and different body parts) are necessary for IR, as well as the extent to which the signals can be manipulated (extra pattern added or image inverted) without affecting IR. The second aim was to test whether the mechanism of face recognition differs from that for non-face objects using the classic face-inversion paradigm and the accuracy of discriminating faces and non-face objects. We used both ecologically realistic settings (mating test) and a conditioned test (electric shock two-alternative forced-choice [TAFC] design) to assess strict IR in medaka. Understanding the cues that animals use to recognise others, as well as their cognitive basis, can help us to elucidate how animals connect to each other in their social world.

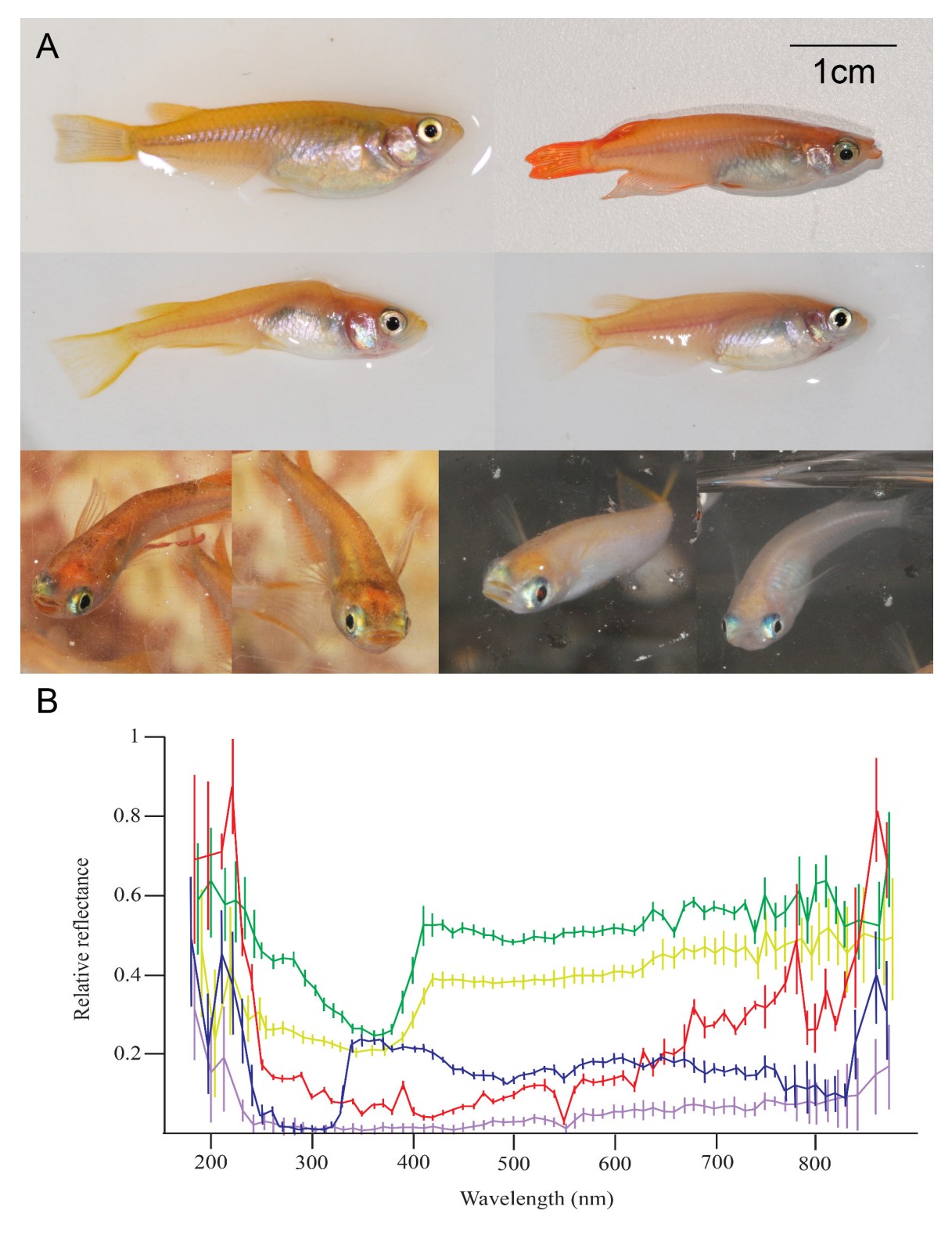

**Figure 1.** Morphological differences between individual medaka fish. (**A**) Medaka individuals may differ in pattern, colour or body shape. The colour and pattern may change based on lighting conditions, physiological conditions and stress level. (**B**) Mean ± SEM relative reflectance of fish body trunks from five individuals from *Figure 1A*. Each colour represents one individual fish. Even though the fish look similar under human vision, their reflectance spectra can be very different.

## Results

### Visual cues are sufficient for medaka individual recognition

First, we tested cues from different modalities to determine which cue is important for medaka IR, using a mating paradigm in which females more quickly accept familiar males. We exposed females to cues from males through different modalities (visual, odour, both visual and odour, and no cue) for at least 5 hr, and placed the pair of fish together for a mating test. Female medaka took significantly less time to accept a male when familiarised with his visual cues or with both his visual and odour cues before mating, compared with an unfamiliar male (ANOVA, $F_{3,76}=5.35$, p=0.002; Tukey's HSD, p<0.05; *Figure 2A*). When different males were substituted in after visual familiarisation, the females were able to recognise the difference and required more time to accept the

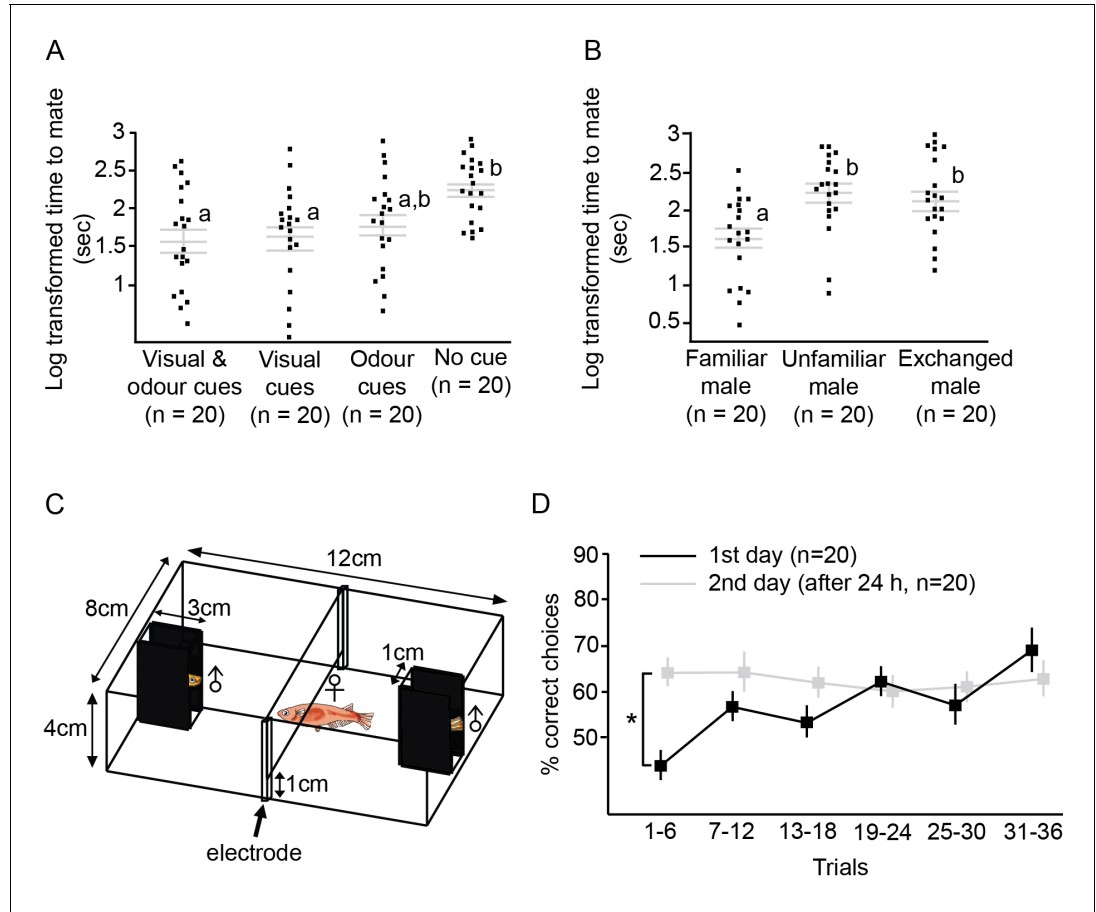

**Figure 2.** Mating test and electric shock two-alternative forced-choice (TAFC) test were used to examine medaka individual recognition (IR). (A) Females were familiarised with different types of male cues for more than 5 hr and then the males and females were placed together for mating tests. Grey lines indicate log transformed mean ± SEM time for females to mate. Different letters indicate statistically significant differences after a Tukey's post hoc test (p<0.05). Each dot represents an individual female. With visual cues alone, the females were able to accept males as familiar mates and required less time to mate. (B) Log transformed time to mate for familiar males (females familiarised with visual cues), unfamiliar males (females given no cue), and exchanged males (females familarised with visual cues from a different male). After substituting the males, the females were able to detect the change and required more time to accept the males. (C) Setup of the electric shock TAFC experiment. The side views of the males were covered. Females were allowed to choose between two unfamiliar males, and when the female entered the area containing the 'incorrect' male, she was given an electric shock. When the female remained in the 'correct' side for more than 3 min, it was considered that she had made a correct choice, and no shock was given. (D) We tested whether medaka females could discriminate different males with the electric shock-conditioned test. The figure shows the mean ± SEM percentage of correct choices in the electric shock task for two consecutive days. Females were able to distinguish individual males associated with electric shock and performance was improved in the last six trials on the first day. Even after 24 hr, the females could still remember the males and made significantly more correct choices than in the first six trials on the first day.

substituted male than a familiar male ($F_{3,57}$=6.49, p=0.003; Tukey's HSD, p<0.05; *Figure 2B*). Females were also able to discriminate between individual males by conditioning with electric shock. We used a TAFC design in which two unfamiliar males were placed at two ends of the setup, and the female was given an electric shock when she entered the side containing the 'incorrect male' (*Figure 2C*). In the last six trials of the experiment, females made significantly more correct choices than in the first six trials (paired *t*-test, $t_{38}$=4.68, p<0.0001; *Figure 2D*). After 24 hr, females were able to discriminate the males and made significantly more correct choices than on the first day during the first six trials (paired *t*-test, $t_{38}$=5.35, p<0.0001 *Figure 2D*).

## Latency for visual familiarisation

We tested how much time female medaka required to accept a male as a familiar mate, and how long the effect lasted. We visually familiarised male medaka with females for 1, 2, 3, and >5 hr, and then put them together for a mating test. We also visually familiarised pairs of medaka for more than 5 hr, and separated them for 1, 2, 3, and 24 hr before the mating test. After visual familiarisation for 3 hr, the females accepted the familiarised males significantly faster than the unfamiliar

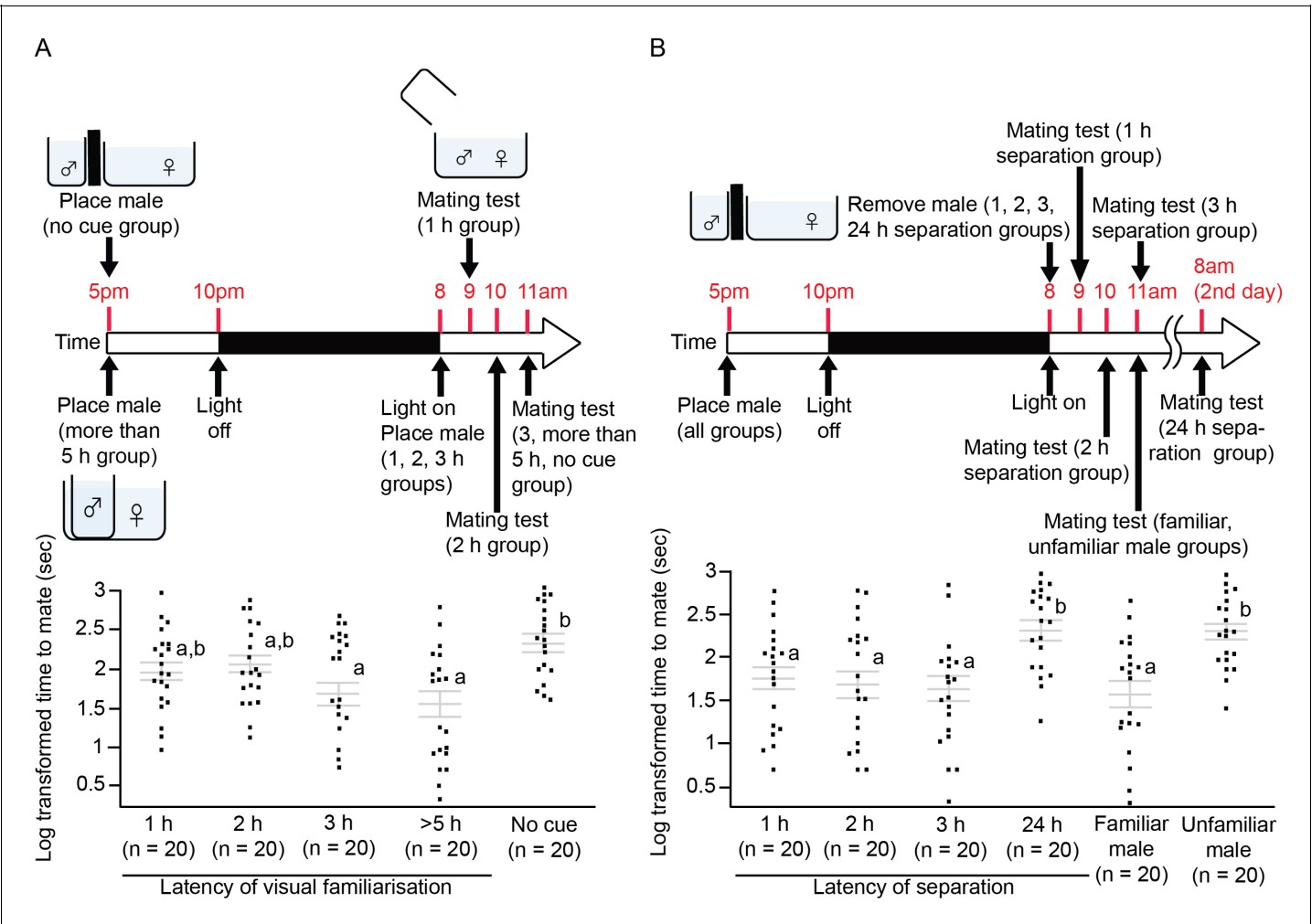

**Figure 3.** Illustration of the experimental protocol and the time required for female medaka to mate. Grey lines indicate log-transformed mean ± SEM time required for females to mate with different groups of visually familiarised males. Letters represent significant differences after analysis of variance tests (ANOVAs) and Tukey's post hoc tests. Dots indicate individual fish. (**A**) Female medaka were visually familiarised with a male for different durations. The effect of visual familiarisation was significant after 3 hr of habituation. (**B**) Pairs of medaka were separated for different durations after being visually familiarised for >5 hr. Even after separation for 3 hr, the females still treated the males as familiar mates; this was no longer the case after 24 hr.

males ($F_{4,95}$=5.39, p=0.0006; Tukey's HSD, p<0.05; *Figure 3A*), and the effect lasted for at least 3 hr after separation, but not for 24 hr ($F_{5,113}$=6.84, p<0.0001; Tukey's HSD, p<0.05; *Figure 3B*). Thus, the outcome differed from that of the electric shock experiment, in which female fish remembered an individual male even after 24 hr.

## Signals around the head may be important for medaka IR

We examined whether motion was involved in medaka IR, and we also looked at the importance of cues from different body parts. We familiarised female medaka to the movements of male medaka using semi-transparent films to obscure their appearance but not their movements. The response of females to the motion-familiarised males did not differ significantly from the response to unfamiliar males (*t*-test, $t_{28}$=−0.03, p=0.97). Familiarisation with only the appearance and not the motion of the males (males fixed in a transparent container) was sufficient for the females to require significantly less time to accept the males ($F_{4,70}$=3.85, p=0.007; Tukey's HSD, p<0.05; *Figure 4A*); however, females required significantly more time to accept head-covered males than tail-covered males (*t*-test, $t_{32}$=−2.33, p=0.03; *Figure 4B*). Even when black spots were added to the faces of the males after visual familiarisation, females still accepted the males as familiar mates ($F_{2,42}$=0.22, p=0.80; *Figure 4C*).

## Medaka failed to recognise inverted faces

We tested whether medaka can recognise inverted faces using both mating tests (*Figure 5A*) and electric shock TAFC tasks. In the mating tests, the time to mate was significantly longer for females familiarised with the vertically flipped images of the males, compared with those familiarised with horizontally flipped and upright images ($F_{2,42}$=5.00, p=0.01; Tukey's HSD, p<0.05; *Figure 4D*). We tested the face inversion effect with the electric shock TAFC tasks as well, also using non-face objects as a control. Fish were trained to discriminate between two individuals or between two sets of non-face objects that differed in familiarity (*Figure 5B*). The fish were exposed to the familiar non-face objects from hatching. Fish were able to discriminate between two fish, two non-face objects, and two familiar non-face objects. They made significantly more correct choices (pair t test, fish: $t_{38}$=2.87, p=0.007; non-face objects: $t_{38}$=3.09, p=0.004; non-face familiar objects: $t_{18}$=2.72, p=0.014) for fish/object presented in the upright position for the last six trials (mean ± standard deviation percentage correct choices, fish: 57.50 ± 16.64; non-face objects: 54.17 ± 16.99; familiar objects: 61.67 ± 17.67) than for those in the first six trials (fish: 44.17 ± 12.49; non-face objects: 39.17 ± 13.55; familiar objects: 40.00 ± 17.92). We examined the effects of visual stimuli type and stimuli orientation on the percentage of correct choices using two-way ANOVA (*Figure 5C*). There was a significant interaction between stimulus type and orientation ($F_{2,94}$=3.68, p=0.03). Therefore, simple main-effect analysis for stimulus type was performed with a Bonferroni adjustment. All pairwise comparisons were run for each simple main effect. In the fish discrimination group, the correct choices decreased significantly after the image was inverted ($F_{1,94}$=12.26, p=0.001), but this was not the cases when sets of two objects were used as stimuli (non-face objects: $F_{1,94}$=0.25, p=0.62; familiar non-face objects: $F_{1,94}$=0.22, p=0.64). There was a significant difference in correct choices between the three types of upright stimuli ($F_{2,94}$=4.68, p=0.01). The correct choices were significantly more frequent for the upright fish stimuli compared to the upright non-face objects (p=0.01), but not for familiar objects (p=0.24). There was no significant difference between two sets of non-face objects (p=1.0).

## Discussion

We demonstrate here that medaka fish are able to perform strict IR in both an ecologically relevant paradigm (mating test) and a conditioning setting (electric shock test). IR is a complex form of recognition and may require strong evolutionary force. For example, nesting penguins use simpler parameters for parent–chick recognition than do non-nesting species (*Jouventin and Aubin, 2002*). Without nest-site information, non-nesting penguins may face higher selection pressure for specific IR ability. Wild medaka are frequently observed in high-density groups (more than hundreds in one pond, Wang and Takeuchi, personal observation) and without obvious, constant nest site or territory. Also, medaka spawn every day and appear to have complex social interactions such as courtship (*Walter and Hamilton, 1970*), mate-guarding behaviour (*Weir et al., 2011*; *Yokoi et al., 2015*),

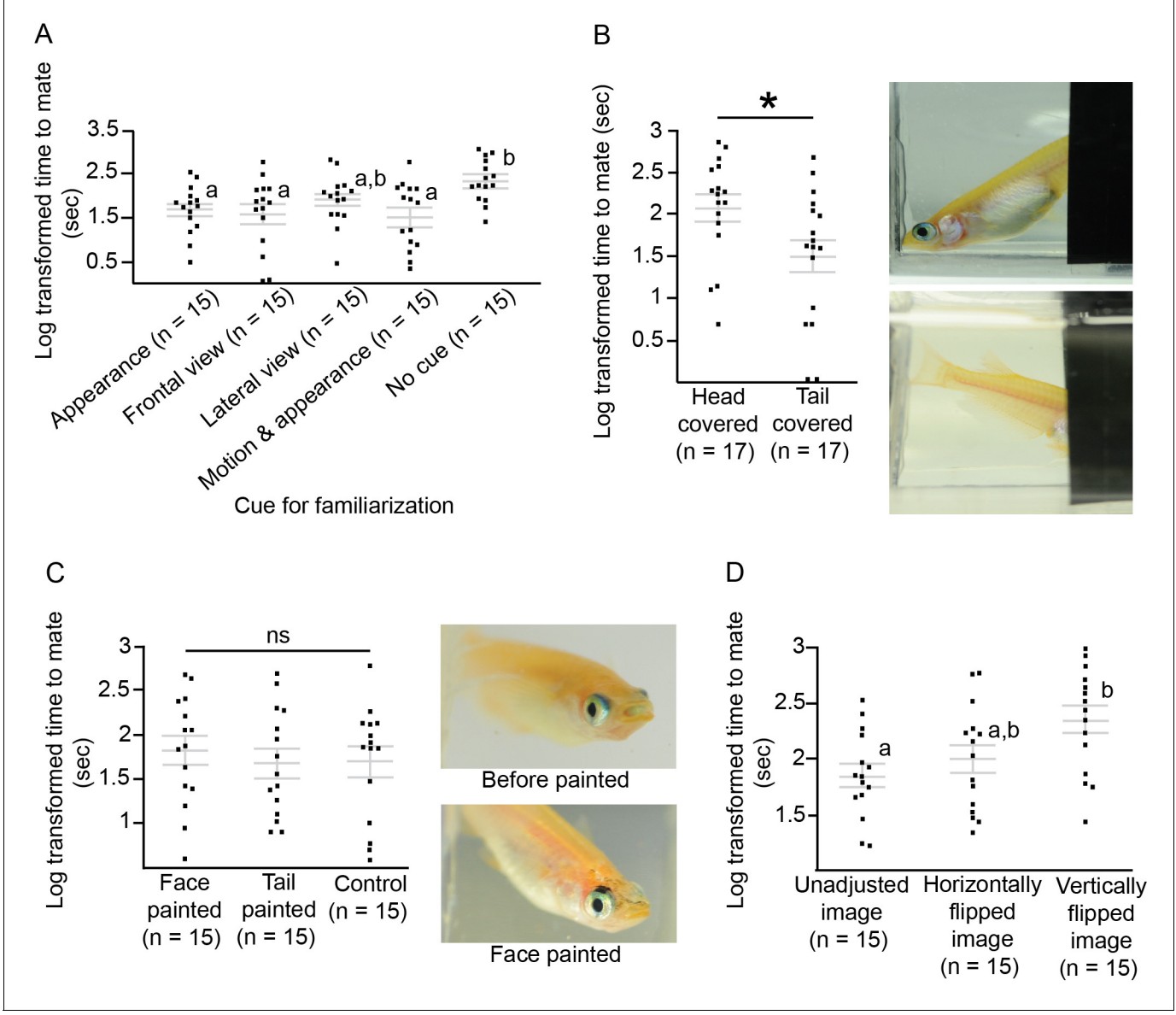

**Figure 4.** Which morphological traits are important for medaka individual recognition, and how they can be modified. Grey lines indicate log transformed mean ± SEM time to mate with different groups of visually familiarised males. Dots indicate individual fish and asterisks indicate p<0.05. Letters represent significant differences after ANOVA and Tukey's post hoc tests. (A) Females were visually familiarised with different types of male visual cues, including appearance and motion. Females were able to recognise the males as familiar mates on the basis of appearance alone. (B) When the head of the male medaka was covered, females were not able to recognise the familiar male and the time to mate was increased. Photos show head-covered and tail-covered medaka. (C) Signals proximate to the head are important for medaka individual recognition. Females were still able to recognise the males after extra spots were painted on the faces of males after visual familiarisation. In the control group, the males were painted by brush with no ink on the face. Photos show one medaka before and after black ink was painted on the face. (D) Images of the males were manipulated with a prism during visual familiarisation, which was followed by mating tests. When familiarised with vertically shifted images, females did not treat the males as familiar mates.

dynamic group-forming and social learning (*Ochiai et al., 2013*). Such frequent and repeated social interactions may induce strict IR (*Tibbetts, 2004*) and sophisticated cognitive/neural adaptation. Medaka and related species provide an excellent model for investigating how the identity signals and recognition ability has evolved, and how animals link multiple identity signals to different social interactions. They are also widely used as genetic and developmental models for social interaction;

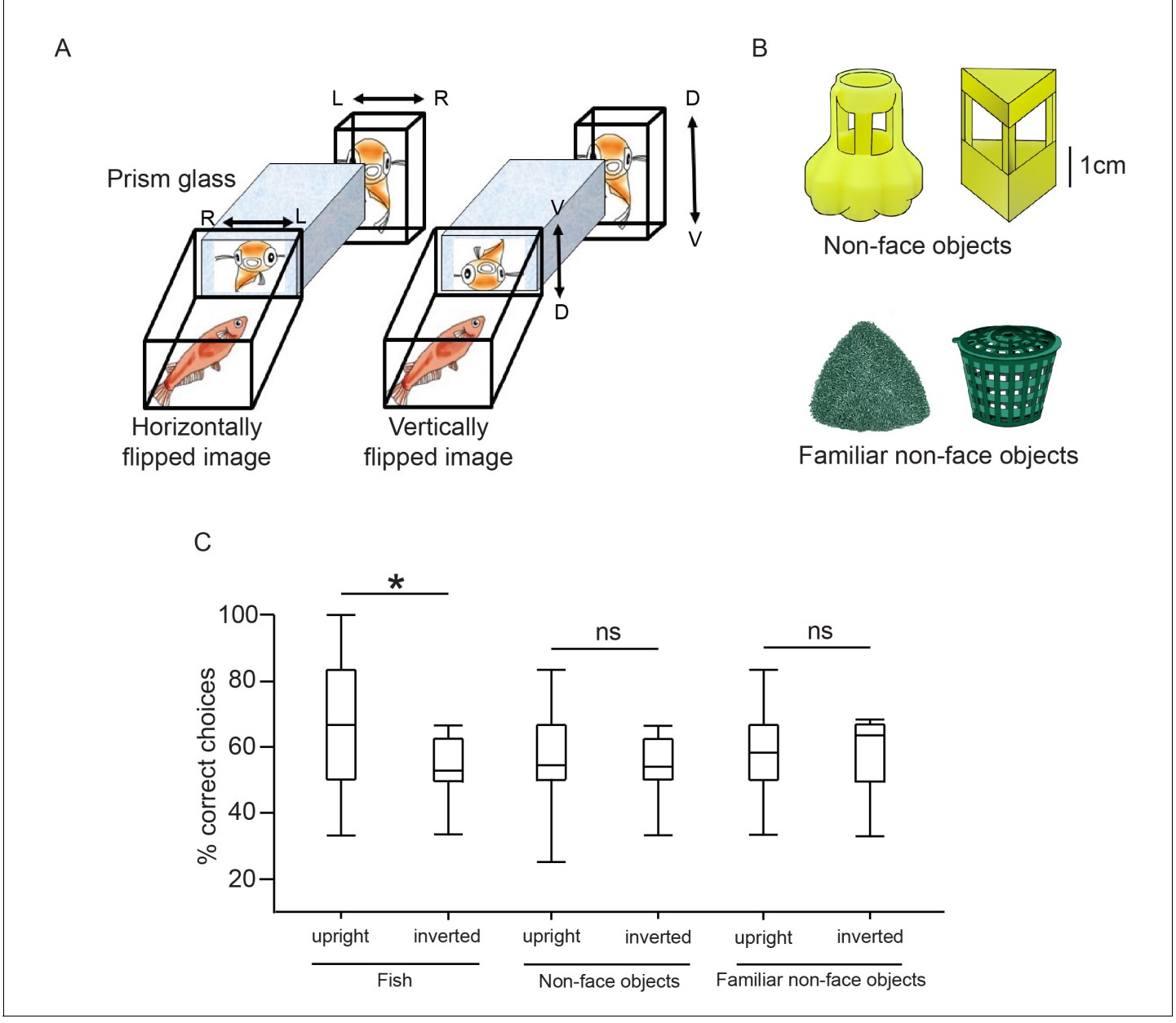

**Figure 5.** We tested how medaka fish recognise inverted fish and objects. (A) Setup for the prism glass test. L, left; R, right; D, dorsal; V, ventral. (B) Two sets of non-face-object stimuli were used in the electric shock two-alternative forced-choice (TAFC) tasks. The fish had been exposed to the familiar objects since hatching. (C) Box plots of percentage correct choices from 6 trials before and after the signals were inverted in the TAFC tasks. Fish were trained to discriminate between two fish or two sets of non-face objects for discrete 36 trials, in addition to 6 inverted trials. The ends of the whiskers represent the minimum and maximum of all of the data.

for example, TN-GnRH3 neurons function as a gateway for activating mate preferences (*Okuyama et al., 2014*), but we do not yet know whether these neurons regulate sensory perception or the decision-making process after signals are perceived. Here, we tested how medaka fish link identity signals to mating partner or conditioned punishment, both of which are rarely described in animal IR literature. Even though medaka are not monogamous, they still have an astonishing ability to recognise mates, suggesting that IR in mating system may be more common than previously thought.

Medaka can successfully differentiate individuals using visual cues alone. More specifically, they use signals around the face for IR. Few animals other than mammals use faces to discriminate individuals, and these species are considered to use relatively simple mechanisms to encode facial features. Two species of cichlid fish use the face to recognise shoal mates or mating partners, and when the facial patterns are exchanged with those of other individuals using digital models, the recognition was found to be based on facial features alone (*Kohda et al., 2015*; *Satoh et al., 2016*). A species of wasp uses a number of facial spots to rank dominance, and this ranking can be artificially altered by adding extra patterns (*Tibbetts and Dale, 2004*). In our study, even after spots were painted onto the faces of the male medaka, the females still treated the male as a familiar mate. This suggests that medaka are able to tolerant some level of local change during IR. More interestingly, medaka showed the classical face-inversion effect, with fish taking a longer time or failing to recognise the inverted faces, but not the inverted non-face objects. To the best of our knowledge, this is the first study to explore the face-inversion effect in animals other than mammals. The inversion effect is indirect evidence for configural/holistic face processing and has been found in humans (*Maurer et al., 2002*), chimpanzees (*Parr et al., 1998*) and sheep (*Kendrick et al., 1996*). These animals not only perceive faces by using internal features, but also make use of configural cues which combine the sum of a number of parts (*Diamond and Carey, 1986*; *Peirce et al., 2000*; *Tomonaga, 2007*; *McKone and Yovel, 2009*). When the faces are upside-down, this configural recognition is impaired, and discrimination times and accuracy deteriorate. The configural recognition process is generally unique to faces and does not appear in other stimuli, although a few special cases have been reported (*Diamond and Carey, 1986*). In addition, specialised neural systems are found to encode faces in humans and some other mammals. In humans, inverted faces delay the neural correlates of faces and increase the activity in object-processing areas (*Aguirre et al., 1999*; *Haxby et al., 1999*). We do not know whether medaka have specific face processing pathways, but regardless of whether medaka sharecommon mechanisms with mammals, these fish can be an important comparative model. Dorsal parts of the telencephalon (pallium) in teleost fish are hypothesised to be related to the mammalian cerebral cortex, including the hippocampus and the pallial amygdala (*O'Connell and Hofmann, 2012*), and thus could be a possible candidate brain region for face-recognition processing.

It is worth noting that the face inversion effect is not direct evidence for holistic processing (*Valentine, 1988*). Further experiments such as the composite task (*Young et al., 1987*), the part-whole task (*Tanaka and Farah, 1993*) and the part-in-spacing-changed-whole task (*Tanaka and Sengco, 1997*) are necessary to test whether the animal uses holistic cues to process faces. But at least, we show the possibility that the mechanism for detecting faces may be different from that for other stimuli. Other hypotheses should also take into consideration, such as the within-class discrimination (*Damasio et al., 1982*) or the expertise hypotheses (*Diamond and Carey, 1986*). The within-class discrimination hypothesis proposes that the special property of faces is due to individual-level discrimination within one type of stimuli, which is a relatively difficult task. For example, discriminating between individual dogs is more difficult than discriminating a dog from a set of mammals such as cats, sheep and monkeys. Although the hypothesis is mostly rejected in humans, it is still possible that it applies to other animals. Another ongoing debate is that humans are face experts and generally use facial stimuli more often than other objects, so the face-specific mechanism is actually expertise-specific. Here, we tested a pair of familiar non-face objects to which the fish had been exposed constantly since they had hatched, in order to control the familiarity level with fish faces. The medaka had a similar level of accuracy when discriminating between familiar objects or medaka faces in the upright orientation, which shows that the familiar objects are sufficient to control for the familiarity and task difficulty in the inversion experiment. The accuracy of discrimination was significantly lower for non-familiar objects compared to that for faces, but we do not know whether the difference was due to task difficulty or familiarity level. In humans, our ability to match unfamiliar faces is surprisingly low. More studies are necessary to understand how familiarity level influences medaka IR, and under which circumstances they can perform IR. One study demonstrated that medaka failed to discriminate fish from their own strain under monochromatic light, whereas they showed strong preference for same-strain mates under normal lighting conditions (*Utagawa et al., 2016*). The males from both strains were familiar to the females, so colours may be important for identifying between strains. We do not know whether medaka show the face inversion effect

for IR under monochromatic light as do humans and monkeys (*Dittrich, 1990*; *Yovel and Duchaine, 2006*; *McKone and Yovel, 2009*).

When being successfully recognised by others is favoured by selection, distinctive traits among individuals may evolve to increase the possibility of being identified (*Tibbetts and Dale, 2007*). Even though medaka may distinguish each other with visual cues, we cannot find obvious visual traits that vary substantially between individuals. One possible explanation for this is that medaka have eight types of cone opsins, and maximum wavelength absorbance ranging from 356 nm to 562 nm (*Matsumoto et al., 2006*), whereas human vision has just three types of opsins with absorbance ranging from 430 nm to 560 nm. Although difficult to detect for human eyes, there is some level of individual difference in reflectance spectra from medaka bodies (*Figure 1B*) and craniofacial morphology (*Kimura et al., 2007*). On the other hand, even individuals in a group are not especially easy to discriminate; when the signal receivers can benefit from successful individuation, the ability for IR can be favoured by selection (*Johnstone, 1997*; *Dale et al., 2001*). Specific identification abilities, such as face recognition, may also evolve under strong selection pressure. Another possible explanation is that medaka do not have to link many individuals with fitness-related tasks, or do not have to remember the individual for long time, which may decrease the resources necessary for IR. Distinguishing faces from other species can also be difficult. For example, sheep can discriminate sheep faces with only 5–10% differences (*Tate et al., 2006*), which is difficult for non-experienced humans. Thus, arguing that medaka lack individual-level variation from human's point of view may be inappropriate. Future research should look at whether medaka can link one or more individuals to multiple ecological tasks, and at whether they can connect identity signals to other fitness-related information such as mate quality or health condition. Rich inbreed lines that are available in medaka could also be useful tools for investigating the heritability of identity signal recognition. For example, in human twin studies, face discrimination ability is heritable for upright faces, but not for inverted faces or other objects (*Mckone and Coltheart, 2010*).

In the mating test, females accepted familiar males after 3 hr of separation, but not after 24 hr. Even after 24 hr, however, females were able to discriminate between males in the electric shock experiment. One possibility is that the females were able to remember the males, but chose not to accept them. This demonstrates that the mate preferences of medaka females are influenced by familiarity level, but with some flexibility. Another explanation is that repeated conditioning strengthens the memory formation, but as yet there is no evidence related to learning ability in either setting. Some other fish species also prefer familiar mates (*Korner et al., 1999*; *Sogabe, 2011*; *Boyle and Tricas, 2014*), and the preference for familiar individuals has been reported to be formed after 4 min (*Dugatkin and Alfieri, 1991*) or 12 days (*Griffiths and Magurran, 1997*) and can last for 2 months (*Brown and Smith, 1994*), depending on the ecological necessity. In medaka, mate-guarding behaviour by males before and after mating has been reported (*Weir et al., 2011*; *Yokoi et al., 2015*) and dominant males are able to remain close to the females, which is a possible explanation for such an ephemeral female preference. Medaka females can spawn daily; thus, flexibility in mate preference can ensure mating with the strongest male at the time. In addition, many types of parasitic mating are reported in medaka (*Koya et al., 2013*), which could facilitate strict IR for recognising the correct mate. Although humans and chimpanzees can differentiate faces immediately (*Parr et al., 2000*), other animal models generally require a longer period for conditioning identity signals to reward or punishment. For example, sheep require at least 30 to 40 trials to condition an unfamiliar face to food reward (*Kendrick et al., 1996*). With similar numbers of trials, medaka are able to condition one individual with electric shock punishment. It should be taken into consideration that almost all animal IR experiments are linked with an ecologically relevant task (e.g. mating in this study) or a conditioning test (either positive or negative conditioning). Other unavoidable factors such as the physiological condition or stress level of the animals can influence recognition. For example, mating is a sensitive and bipolar procedure which is influenced by the condition and interaction of both individuals. Multiple paradigms with suitable controls may be useful to assess the evidence of convergent IR in animals.

Overall, our data suggest that medaka can perform strict IR and that, as in humans, specific visual features such as the face may be more important for IR than others. Medaka also show the classic face-inversion effect, which could indicate that specific processes are involved in recognising faces. It is likely that the mechanism underlying medaka face recognition differs from that in mammals. The application of the rich genetic toolset available for this species, which includes genome

editing tools (such as CRISPR/Cas9) and epigenetic methods, will allow more detailed investigation of the specialised cognitive abilities (*Ansai and Kinoshita, 2014*; *Nakamura et al., 2014*). Even a brain as small as that of the medaka is able to manage such a complex cognitive task. A understanding of how faces are perceptually encoded in simpler models will provide concepts that may be exploited both for the development of new machine face-recognition systems and to explain how the brain processes highly homogenous social information in general. The advantages and limitations of this model compared to mammalian models in face recognition will allow interesting future investigations of convergent systems from phylogenetically distant groups. Other than looking to provide a comparative view on the neurobiology of faces, our future direction will focus on IR in the real world. For example, we would like to study how medaka link individuals to multiple ecological-related topics and how IR shapes their societies and group forming (*Wang et al., 2015*). The evidence gathered in such studies will indicate the evolutionary background in which such sophisticated cognitive process were formed, which is important for all social animals.

# Materials and methods

## Study animals and ethics statement

A total of 569 adult medaka fish (*Oryzias latipes*, drR strain), aged 6–18 months, were tested in this study. Fish were maintained as described in *Okuyama et al. (2014)*. The animal experiments were performed as approved by the Animal Care and Use Committee of the University of Tokyo (permit number: 12–07). All efforts were made to minimise suffering according to the NIH Guide for the Care and Use of Laboratory Animals.

## Mating test

### General protocol

Prior to the experiment, male and female medaka were randomly paired and housed in experimental tanks (19 cm × 13 cm × 12 cm). Once egg production was observed for more than three consecutive days, the fish were used for experiments. Fish were tested in their home tanks following the previously described protocol (*Okuyama et al., 2014*). Fish were moved to a shelf 10 cm away from the original position and recorded from above with Nikon d300s and d90 cameras. The time to mate was taken as the time from the first courtship display (circle dance) to the cross (male and female cross their bodies) followed by spawning. Sample size was calculated from a pilot study (*Okuyama et al., 2014*) with 0.8 power at a two-tailed significance level of $p=0.05$ using SPSS v.22 (SPSS Inc., Chicago, IL, USA). There was no significant outlier and all of the data points were included. The time to mate was log transformed to satisfy the assumption of normality and analysed by two-tailed one-way ANOVA followed by Tukey's post hoc test.

### Multimodal recognition test

To test whether female medaka use multiple modalities to discriminate familiar males, we randomly assigned 80 females to be exposed to (1) a visual cue; (2) an olfactory cue; (3) both visual and olfactory cues; and (4) no cue of male medaka for >5 hr. In the visual familiarisation group, a male was held in a transparent glass cup (9 cm high, 7 cm in diameter) through which the male and female could see each other before the male was released to the tank for the mating test. In the olfactory familiarisation group, the glass was opaque with small holes through which water could pass, but the female could not see the male inside. In the visual and olfactory familiarisation group, the glass was transparent with holes, and in the control group, a transparent glass was placed in the home tank and a male was placed in the glass cup outside the female's sight. Mating experiments were performed on the following day between 09.00 and 11.00.

### Male exchanged visual recognition test

To determine whether female medaka could visually identify a particular male, we visually familiarised 20 females with males, but then substituted an unfamiliar male for the mating test. Another 40 pairs of medaka were randomly assigned to visual familiarise task and no cue task as described above.

## Latency necessary for visual familiarisation

We tested the amount of time the females required for visual familiarisation and how long they could be separated from the familiarised males but still recognise them as familiar. Male medaka (n=100) were randomly assigned to be visually familiarised with females for 0, 1, 2, 3, and ≥5 hr before the mating test. An additional 100 females were visually familiarised with males for >5 hr and separated for 0, 1, 2, 3, and 24 hr before the mating test. As a control, 20 females were tested with unfamiliar males.

## Assessing motion and appearance cues for visual recognition

To assess whether fish use motion or appearance cues for visual recognition, we familiarised 15 pairs of medaka to the motion of the male, separating the individuals within a pair by a semi-transparent film for 3 hr, during which the females could observe the movement of the male but not his appearance. An extra light source was set at the side of the male medaka to facilitate the projection of his shadow onto the film. In the control group, 15 males were placed out of sight of the females. In the appearance test, the males were first habituated in transparent plastic containers (3 cm × 1 cm × 4 cm) which they cannot move freely for 3 hr each day and for at least 5 days. In the main experiment, 15 males were placed in the transparent plastic containers (3 cm × 1 cm × 4 cm) and the containers were placed inside the females' home tanks for visual familiarisation. Females could swim freely in the tank and observe the males. After visual familiarisation for 3 hr, the males were released for the mating test. To investigate the body part used for visual recognition, 94 males were placed in the same experimental arrangement as in the previous experiment, and during each visual familiarisation, either the lateral side (n=15), frontal side (n=15), head or tail (n=17 each) was covered with a black plastic board so as to be out of sight of the females. As controls, 15 males were held in a transparent glass cup (9 cm high, 7 cm in diameter) so that the females could see both the appearance and motion of the males, and another 15 females were tested with unfamiliar males.

## Face-painted test

We painted a pattern on a male medaka's face or tail (n = 15 each) using a black marker after visual familiarisation with a female for more than 5 hr. The manipulation required less than 10 min, and the pair of fish were then placed in the same tank for the mating test. The males in the control group (n = 15) were painted on the face using a brush with no ink.

## Prism test

We inverted the image of a male medaka horizontally or vertically using a prism (9 cm × 3.5 cm × 2.5 cm) during 3 hr of visual familiarisation, followed by a mating test to investigate the females' ability to recognise the male. A normal glass was used in the control group (n = 15 in each group).

## Electric shock experiment

### General protocol

According to the previously described study design (*Blank et al., 2009*), experiments were performed in aquaria (12 cm × 8 cm × 4 cm) divided into two parts with a transparent divider in the middle which allows fish to pass underneath. The surrounding walls of the aquaria were covered with cork sheets to prevent reflection. We used a TAFC design with signals at the opposite side of the aquaria. The test was initiated by introducing the female into the centre of the experimental arrangement, and a correct choice was determined by the female remaining on the side of the 'correct signal' for >3 min. Once the female entered the side of the 'incorrect signal', a 4 V, 0.5 s electric shock was administered as 'punishment' (custom made by Extion Co. Ltd., Taipei, Taiwan). The positions of the stimuli were changed randomly. Two-tailed paired *t*-tests were used to compare the change in performance between different trials. Before the experiment, fish were tested with a black and white signal pair following the design described by *Blank et al. (2009)*. Only those fish that made five consecutive correct choices (correct colour was assigned randomly for each fish) were used for the subsequent experiments.

### Fish discrimination test

To assess whether female medaka could be conditioned to individual males by electric shock, we placed two unfamiliar males into transparent plastic containers (3 cm × 1 cm × 4 cm) placed at each end of the apparatus as stimuli. The side views of the males were covered so that the female could only see the front view of the males (*Figure 2C*). One male was randomly selected as the 'incorrect' choice for the female, with the female being conditioned with an electric shock. Twenty females were tested for 36 discrete trials. An additional six trials were performed with the face of the males vertically shifted by prisms. After 24 hr, the same experiment was repeated except inversion to test the females' memory.

### Non-face object discrimination test

To test whether medaka can distinguish fish faces faster than non-face objects, and whether they can immediately recognise inverted objects, we used two sets of objects differing in familiarity levels for electric shock TAFC tests (*Figure 5b*). Twenty fish were electric shock conditioned with the non-face objects for 36 discrete trials, and the shocked objects were randomised. The fish were tested in an additional six trials with the same objects shifted upside-down. In order to control for the familiarity level, another 10 fish were tested for two familiar objects to which they had been exposed since hatching (*Figure 5b*), with the same procedures described above. Two weeks before the experiment, the objects were placed in the centre of the tanks to make sure the fish were familiar with all angles of the objects.

## Measurement of medaka reflectance spectra

We sacrificed the fish using a −20° freezer and placed them in a Petri dish for measurement. The reflectance spectra of the body trunks from five medaka were measured by a spectrometer (FLAME-S-UV-VIS-ES, Ocean Optics, Inc. FL, US). A light source (DH-MINI) providing UV to visible light output illuminated the probe (R400-7-SR) under an angle of 45° to the fish trunk. The reflectance spectra of the fish were recorded with a resolution of 1 nm relative to a white standard (WS-1) with OCEAN-VIEW software (Ocean Optics, Inc. FL, US).

## Acknowledgements

We are grateful to NBRP Medaka (https://shigen.nig.ac.jp/medaka/) for providing d-rR/TOKYO (Strain ID: MT837). We would like to thank Prof. Takeo Kubo for providing lab space and support, and OptoSirius Inc. (Tokyo, Japan) for providing optical equipment. We also thank Prof. Larry Young and Dr Teruhiro Okuyama for their comments on the work. MYW is supported by the FY2013 JSPS Postdoctoral Fellowship for Foreign Researchers and JSPS Grant-in-Aid number 25–03074. HT is supported by JSPS KAKENHI Grant Numbers 26290003, the Grant-in-Aid for Scientific Research on Innovative Areas 'Memory dynamism' (26115508) from the Ministry of Education, Culture, Sports, Science, and Technology; the Brain Science Foundation; and the Yamada Science Foundation. The funders had no role in the study design, data collection and analysis, decision to publish, or preparation of the manuscript.

## Additional information

### Funding

| Funder | Grant reference number | Author |
| --- | --- | --- |
| Japan Society for the Promotion of Science | JSPS Grant-in-Aid number 25-03074 | Mu-Yun Wang |
| Ministry of Education, Culture, Sports, Science, and Technology | Grant-in-Aid for Scientific Research on Innovative Areas | Hideaki Takeuchi |
| Brain Science Foundation | | Hideaki Takeuchi |
| Yamada Science Foundation | | Hideaki Takeuchi |
| Japan Society for the Promo- | JSPS KAKENHI 26290003 | Hideaki Takeuchi |

tion of Science

The funders had no role in study design, data collection and interpretation, or the decision to submit the work for publication.

## Author contributions

M-YW, Conceptualization, Resources, Data curation, Software, Formal analysis, Funding acquisition, Validation, Investigation, Visualization, Methodology, Writing—original draft, Project administration, Writing—review and editing; HT, Conceptualization, Resources, Software, Supervision, Funding acquisition, Validation, Investigation, Project administration, Writing—review and editing

## Author ORCIDs

Mu-Yun Wang, http://orcid.org/0000-0002-9533-995X

Hideaki Takeuchi, http://orcid.org/0000-0001-6610-3895

## Ethics

Animal experimentation: The animal experiments were performed as approved by the Animal Care and Use Committee of the University of Tokyo (permit number: 12-07). All efforts were made to minimize suffering according to the NIH Guide for the Care and Use of Laboratory Animals.

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
