## [Decision Letter]

Thank you for submitting your article "Individual recognition and face inversion effect in medaka fish *Oryzias latipes*)" for consideration by *eLife*. Your article has been reviewed by two peer reviewers, and the evaluation has been overseen by a Reviewing Editor and Sabine Kastner as the Senior Editor. The following individuals involved in review of your submission have agreed to reveal their identity: Galit Yovel (Reviewer #2).

The reviewers have discussed the reviews with one another and the Reviewing Editor has drafted this decision to help you prepare a revised submission.

Summary:

The paper examines identity recognition in medaka fish. Both visual and odor cues were used and medaka fish were shown to use visual information for identity recognition even in the absence of odor information. Two tests were used to examine identity recognition, mating preference and electric shock conditioning. Both paradigms indicate that female medaka fish discriminated males based on their faces. Several additional tests demonstrated that it is the face rather than body motion that was used for recognition, and that recognition was invariant to the addition of a black dot to the face. Finally, recognition did transfer to horizontally flipped faces but did not to vertically inverted faces. In contrast, recognition of novel objects was invariant to vertical inversion. The authors concluded that Medaka fish, similar to humans, show a face inversion effect.

This is an elegant study that includes several informative tests to assess whether medaka fish can recognize individuals based on their faces. The two paradigms used show consistent results but also complement each other as the electric shock conditioning generated more long-lasting effects than the mating preference task. The occlusion of the face or body motion is also very informative showing that identification was primarily based on the face. Also, the fact that the black dot did not interfere with recognition suggests that the individual representation is invariant to local changes.

We are less convinced by the comparison between the face and the object stimuli. We assume that the object stimuli are novel to the medaka fish, which may account for the relatively lower performance on the object task. In human studies, the face inversion effect is typically compared to familiar objects and on tasks in which the upright face and object stimuli are matched for performance. Such findings suggest that familiarity and task difficulty may not account for the differences found between faces and objects. This is not the case for the current results, which may be accounted for by task difficulty and/or familiarity rather than differences in processing mechanisms for faces and objects. Thus, a task that employs a familiar object that generates similar performance as the upright face task is needed to examine whether the inversion effect is face specific.

Essential revisions:

1) The authors indicate the electric shock task generated longer lasting effect than the male preference task (Figure 3). To support this claim, it is necessary to perform an ANOVA in which the two tasks are directly compared and a significant interaction between Time x Task is found.

2) Similarly, the comparison between upright and inverted conditions for faces and shapes should be done with a 2x2 ANOVA to examine if there is an interaction between the two factors (It would also be good to show these four conditions in the same figure). However, even if such an effect is found, it may not address the familiarity and task difficulty as alternative explanations, as indicated above.

3) In the Discussion (first paragraph), the authors cite a study that showed that medaka fish failed to recognize faces under monochromatic light and therefore color cues play a role in medaka face recognition. This finding is inconsistent with human face inversion effect, as no face inversion effect was found when faces differ only in color (Yovel & Duchaine, 2006, McKone & Yovel, 2009). This point should be discussed.

4) It will be good to include a speculation on the brain areas that medaka fish may use to complete such an advanced face recognition task.

5) We had a number of concerns about the situation of the results into a broader context. The Introduction of the paper does not set up why any once should care about these results and would benefit from a clear explanation of the power of medaka fish as either a translational model or a comparative model (i.e., used to understand evolutionary phenomena). Further, it is important to provide basic details about medaka anatomy – specifically the sort of individual level variation in structure or appearance – and social environment – that would support the claim for visual individual recognition. This theme of needing more detail is carried further insofar as it is not clear why we should care why these fish might or might not have IR. Inversion literature cited is not sufficient (for example, Thompson 1980 and the whole "Thatcher effect" literature which is present both for monkeys and people is relevant). We are quite familiar with the individual recognition and inversion literature in primates and via that expertise also know that there is a good deal of IR literature in other species that is not cited. While a comprehensive literature review may not be required, the sparsity of the current intro and discussion leaves the reader wondering if the authors are themselves familiar with the literature. For example, a google search revealed an extensive literature in birds. In a similar vein, we found the Discussion to be a rehashing of the results without a broader perspective of why the results are important or how they are situated in a translational, comparative, or evolutionary context.

6) We have concerns about the statistics given the plots in all of the figures. The authors indicate that the data were normally distributed, but by condition they do not appear to be. There seems to be significant clumping around floor and greater variance in some conditions than others. Taken together, it seems possible that ANOVA is not the appropriate statistical technique.

7) We are confused by Figure 5. Isn't chance performance 50% in the two AFC task? Why does performance start at 30%, and only ramp up to chance at the end?

---

## [Author Response]

Essential revisions:

1) The authors indicate the electric shock task generated longer lasting effect than the male preference task (Figure 3). To support this claim, it is necessary to perform an ANOVA in which the two tasks are directly compared and a significant interaction between Time x Task is found.

Unfortunately because of the nature of the data, we cannot compare the two experiments directly in an ANOVA test. The electric shock task looked at the percentage of correct choices, while male preference task examined the time until mating. We designed the relevant control groups in both experiments, and we think the design was applicable to test how long the effect of familiarity can last.

2) Similarly, the comparison between upright and inverted conditions for faces and shapes should be done with a 2x2 ANOVA to examine if there is an interaction between the two factors (It would also be good to show these four conditions in the same figure). However, even if such an effect is found, it may not address the familiarity and task difficulty as alternative explanations, as indicated above.

Thank you for the useful suggestions. We added a familiar object control group in which the objects were placed in the fish tanks since the fish were born (thus the familiarity level was the same as fish faces). To ensure the fish were familiar with all the angles of the objects, we placed the objects in the centre of the tank two weeks prior to the experiment. In the upright position, fish could discriminate the objects equally well compared with faces (no significant difference between correct choices made by fish when discriminating between two faces or two familiar objects), suggesting the familiar objects were appropriate controls. We compared the faces and two sets of objects in the upright and inverted conditions with a 3x2 ANOVA and found there was a significant interaction between factors. The results were now presented in the same figure (Figure 5).

3) In the Discussion (first paragraph), the authors cite a study that showed that medaka fish failed to recognize faces under monochromatic light and therefore color cues play a role in medaka face recognition. This finding is inconsistent with human face inversion effect, as no face inversion effect was found when faces differ only in color (Yovel & Duchaine, 2006, McKone & Yovel, 2009). This point should be discussed.

We are sorry that this was not clear – Utagawa et al. demonstrated that medaka failed to discriminate fish from their own strain under monochromaticlight, when they showed strong preference for same strain mate under normal lighting condition. Both the males from two different strains were familiar to the females. We do not know whether medaka can perform IR for face inversion effect under monochromatic light. This point is discussed in the third paragraph of the Discussion.

4) It will be good to include a speculation on the brain areas that medaka fish may use to complete such an advanced face recognition task.

We hypothesize that the dorsal parts of the telencephalon (pallium) in medaka may be a possible candidate to process specific recognition tasks such as face recognition (Discussion, second paragraph).

5) We had a number of concerns about the situation of the results into a broader context. The Introduction of the paper does not set up why any once should care about these results and would benefit from a clear explanation of the power of medaka fish as either a translational model or a comparative model (i.e., used to understand evolutionary phenomena). Further, it is important to provide basic details about medaka anatomy – specifically the sort of individual level variation in structure or appearance – and social environment – that would support the claim for visual individual recognition. This theme of needing more detail is carried further insofar as it is not clear why we should care why these fish might or might not have IR. Inversion literature cited is not sufficient (for example, Thompson 1980 and the whole "Thatcher effect" literature which is present both for monkeys and people is relevant). We are quite familiar with the individual recognition and inversion literature in primates and via that expertise also know that there is a good deal of IR literature in other species that is not cited. While a comprehensive literature review may not be required, the sparsity of the current intro and discussion leaves the reader wondering if the authors are themselves familiar with the literature. For example, a google search revealed an extensive literature in birds. In a similar vein, we found the Discussion to be a rehashing of the results without a broader perspective of why the results are important or how they are situated in a translational, comparative, or evolutionary context.

Thank you for the valuable comments. We completely rewrote the Introduction and the Discussion to fit them into a broader perspective. Why medaka can be a good model species was briefly described in the Introduction (fourth paragraph) and further explained in the Discussion (first, second and last paragraphs). We added an experiment testing the reflectance spectra of the medaka, and found even difficult to distinguish by a human observer, a certain level of morphological variation exists between medaka individuals (Figure 1). The morphological variation of medaka was discussed in the Discussion, as well as their visual ability (fourth paragraph).

Medaka ecology and social environments were discussed in the fourth paragraph of the Introduction and in the first and fifth paragraphs of the Discussion. A more complete review of the inversion literatures was added (Introduction, second and third paragraphs, Discussion, second and third paragraphs), including the classic Thatcher effect (Introduction, second paragraph), other experiments related to configural face processing (Discussion, third paragraph) and several theoretical challenges proposing faces may not be special. We reconstructed the logic of the manuscript and emphasized the concept of individual recognition in the beginning (Introduction, first paragraph) and the end (Discussion, first paragraph) of the manuscript, before introducing face recognition in human and other animals. Classic IR examples in birds (Introduction, first paragraph and Discussion, first paragraph) and other animals (Introduction, first and third paragraphs) were included.

6) We have concerns about the statistics given the plots in all of the figures. The authors indicate that the data were normally distributed, but by condition they do not appear to be. There seems to be significant clumping around floor and greater variance in some conditions than others. Taken together, it seems possible that ANOVA is not the appropriate statistical technique.

Thank you for pointing out this error. We log-transformed all the data for the mating tests to better fit the assumption of normality and redo all the stats. Normality was assessed using Shapiro-Wilk's normality test and all the data were normally distributed (p >.05).

7) We are confused by Figure 5. Isn't chance performance 50% in the two AFC task? Why does performance start at 30%, and only ramp up to chance at the end?

Only when the fish stayed in the correct side for more than three minutes, was the decision counted as a correct choice. In such a case, the random choice was not 50%. Medaka, by their nature, moves back and forth in the tank. We needed to train them to stay at one side as making a choice. In the end of all three experimental groups (fish, non-face objects and familiar non-face objects), fish made significantly more correct choices than at the beginning, suggesting the fish were able to improve performance and discriminate between two fish or two objects. Since the data from Figure 5 in the previous version overlapped with the new Figure 5, we deleted the figure and describe the data in the text (subsection “Medaka failed to recognise inverted faces”).